physiology/biophysics/bioengineering

neuro-cardiac, co-culture, optical mapping, sympathetic, arrhythmia

**Author for correspondence:**
Rebecca A. B. Burton
e-mail: rebecca.burton@pharm.ox.ac.uk

---

# Combining tissue engineering and optical imaging approaches to explore interactions along the neuro-cardiac axis

Charalampos Sigalas[1,†], Maegan Cremer[1,†],
Annika Winbo[2,3], Samuel J. Bose[1], Jesse L. Ashton[2],
Gil Bub[4], Johanna M. Montgomery[2]
and Rebecca A. B. Burton[1]

[1]Department of Pharmacology, University of Oxford, Oxford, UK
[2]Department of Physiology, University of Auckland, Auckland, New Zealand
[3]Department of Paediatric and Congenital Cardiac Services, Starship Children's Hospital, Auckland, New Zealand
[4]Department of Physiology, McGill University, Montreal, Canada

 CS, 0000-0003-0405-1385; MC, 0000-0001-9420-4802;
SJB, 0000-0003-2922-3075; RABB, 0000-0002-0904-3862

Interactions along the neuro-cardiac axis are being explored with regard to their involvement in cardiac diseases, including catecholaminergic polymorphic ventricular tachycardia, hypertension, atrial fibrillation, long QT syndrome and sudden death in epilepsy. Interrogation of the pathophysiology and pathogenesis of neuro-cardiac diseases in animal models present challenges resulting from species differences, phenotypic variation, developmental effects and limited availability of data relevant at both the tissue and cellular level. By contrast, tissue-engineered models containing cardiomyocytes and peripheral sympathetic and parasympathetic neurons afford characterization of cellular- and tissue-level behaviours while maintaining precise control over developmental conditions, cellular genotype and phenotype. Such approaches are uniquely suited to long-term, high-throughput characterization using optical recording techniques with the potential for increased translational benefit compared to more established techniques. Furthermore, tissue-engineered constructs provide an intermediary between whole animal/tissue experiments and *in silico* models. This paper reviews the advantages of tissue engineering methods of multiple cell types and optical imaging techniques for the characterization of neuro-cardiac diseases.

†Joint first authors.

# 1. Introduction

Neuro-cardiac disease describes the pathophysiological interaction between the nervous system (NS) and cardiovascular system (CVS), with pathological activity in one system resulting in pathological behaviour in the other [1]. Hypertension, catecholaminergic polymorphic ventricular tachycardia (CPVT), long QT syndrome (LQTS), as well as other syncope and seizure disorders have all been proposed as neuro-cardiac diseases [2–6]. CPVT [2], atrial fibrillation [7], LQTS, ventricular tachycardia [7] and atrio-ventricular (AV) block [8] are coincident with seizure presentation in clinical populations [2,4,7,9]. Seizures have also been reported in up to 50% of CPVT patients earlier in life before the presentation of CVPT-related symptoms [2,10]. This observation raises questions about our fundamental understanding of these diseases. For example, in seizures or cardiac arrhythmias in CPVT patients, is the actual triggering event neuronal or cardiac? This is important to know because it could contribute in the effectiveness of current therapeutic strategies including pharmacological approaches and surgical interventions [11]. Therefore, a greater understanding of the mechanisms underlying the pathophysiology of neuro-cardiac diseases is required to improve the diagnosis and treatment of such disorders.

The CVS is under homeostatic control via the autonomic nervous system (ANS). Aberrant sympathetic and parasympathetic NS activity, hyperinnervation and disorganized neuro-cardiac interfaces have been correlated with cardiac arrhythmia [12,13]. Additionally, plasticity in sympathetic and parasympathetic neuronal networks of the heart and developmental interactions may influence neuro-cardiac behaviours [5,14,15]. Furthermore, cell surface characteristics and secretion of growth factors can shape the development, maturation and ultimately the function of cardiomyocytes and neurons [6,16–18]. Pathogenic interactions between the NS and CVS can operate in either direction. For example, NS to CVS pathogenesis was confirmed in a rodent model by Larsen *et al.* for hypertension phenotypes [6], while the potential for CVS to NS pathogenesis has been highlighted by the observation that premature ventricular contractions (PVCs) have the capacity to excite a population of neurons independent of cardiac pacing and haemodynamic forces [19].

Both neurons and cardiomyocytes are excitatory cells expressing similar ion channels and rhythmic behaviours. Mutations that affect ion channels expressed in cardiac cells have the potential to affect the same channels when expressed in neurons [2,20,21]. For instance, genetic autopsies have revealed gain-of-function mutations in sodium channels (Nav1.5) and type 2 ryanodine receptors (RyR2) as well as loss of function in potassium channels ($K_v$7.1, hERG), which are common to both sudden unexplained death in epilepsy (SUDEP) and cardiac arrhythmias [4,14,21–23]. Dysregulation of plasma membrane potentials and intracellular calcium regulation can prolong depolarization, resulting in concomitant seizure and cardiac arrhythmia phenotypes [7,21].

Neuro-cardiac diseases act at all levels from the cellular to whole and multi-organ, and a detailed understanding is required across these scales. Currently, neuro-cardiac interactions are most commonly studied using whole animal models [14,19,24] and tissue-engineered constructs [6,16]. However, the intricate communication between neural and cardiac tissue in the heart could lead to confounding results in whole animal studies of neuro-cardiac diseases [25]. In comparison, tissue culture models enable experimenter control over the complexity of the system [26] and can help to bridge the gap in understanding the mechanisms of pathophysiology between the cellular and organ levels. Furthermore, the planar surface geometry of tissue culture models is uniquely suited to optical actuation and characterization [27–29]. Advances in optical actuation and characterization will enable long-term, high-throughput characterization of neuro-cardiac diseases [30,31].

In this mini review, we will focus on the recent advances in optical imaging of engineered tissue culture methods that allow the study of synergistic effects of neural and cardiac tissue in neuro-cardiac diseases. To illustrate the importance of such methods, we will first discuss how common cellular mechanisms, such as $Ca^{2+}$ or $K^+$ regulation through transmembrane channels, can affect both neural and cardiac tissue in CPVT and LQTS, as examples of diseases that were initially considered solely cardiac.

# 2. Neuro-cardiac physiology and disease

Calcium plays an integral role in neuro-cardiac transmission affecting both cardiomyocyte contractility and neuronal excitability. Calcium dysregulation underlies many cardiovascular pathologies such as hypertrophy [32,33], hypertension [34,35] and arrhythmias [36–38]. In the central nervous system (CNS) and peripheral nervous system (PNS), calcium is involved in neurotransmitter release, generation of excitatory or inhibitory postsynaptic potentials and the induction of different forms of

synaptic plasticity, such as long-term potentiation (LTP) and long-term depression (LTD) in the hippocampus [39,40] or in the stellate and superior cervical sympathetic ganglia [41,42]. Neuronal calcium dysregulation, therefore, has the potential to influence the functional connectivity within neuronal networks of both the CNS and ANS along the neuro-cardiac axis. Taken together, mutations affecting proteins linked to calcium signalling (such as RyR2) provide a substrate for interplay between the NS and the CVS that can lead to both cardiovascular and neurological pathologies.

CPVT is a lethal genetic disease associated with arrhythmogenesis caused by mishandling of intracellular calcium leading to inappropriate calcium release events from the sarcoplasmic reticulum (SR) and aberrant muscle contractions in the heart [36,43–47]. With symptoms including syncope and sudden cardiac death, CPVT is normally first diagnosed in early childhood to young adulthood with a mortality rate reaching 50% by the age of 20 years [48,49]. Aberrant calcium release in cardiomyocytes can lead to abnormal diastolic membrane depolarization that can ultimately impair the cardiac beating rate and cause arrhythmias. The most frequent form of CPVT, CPVT1, is caused by mutations to the cardiac ryanodine receptor RyR2, for which more than 150 mutations have been reported [36,45]. RyR2 is the main calcium release channel involved in calcium-induced calcium release during excitation–contraction (EC) coupling in cardiac cells [50,51]. In 2006, work by Liu *et al.* [52] elegantly demonstrated that delayed afterdepolarizations (DADs), a major risk factor for arrhythmogenesis, may be linked to RyR2 with the CPVT-related mutation R4496C.

CPVT was originally described as an exclusively cardiovascular disease whereby stress or exercise induces elevation of catecholamines that can lead to ventricular tachycardia [48]. Recent research in humans [53] and mouse models carrying RyR2 mutations associated with CPVT [2,14] shows coincidence of CPVT, seizures and SUDEP, suggesting a possible connection between the induction of cardiac arrhythmias and irregular CNS activity. In fact, pharmacologically induced seizures and brainstem spreading depolarization led to fatal cardiac arrhythmias in mice carrying the gain-of-function mutation RyR2-R176Q [14]. These data suggest that neurological events such as epileptic seizures could be linked to long-term effects on the heart. Nevertheless, CPVT can also be manifested as a neuro-cardiac disease in the absence of abrupt CNS activity. Recent work suggests that reciprocal pathogenesis in the PNS and CVS could underlie the generation of arrhythmias in animal models of cardiac disease [19] and in CPVT patients. For example, cardiac pacing alone is insufficient to induce arrhythmic events in CPVT patients, but when combined with sympathetic stimulation may result in arrhythmia [24]. These results were corroborated in mice expressing the RyR2 mutation R2474S, which exhibit an exercise-induced ventricular tachycardia phenotype with greater sensitivity to β-adrenergic receptor stimulation than cardiac pacing [24].

LQTS is an inherited channelopathy associated with syncope and sudden death in the young, with the two most common LQTS subtypes, LQT1 and LQT2 caused by mutations in genes encoding the potassium channels *KCNQ1* (responsible for the slow delayed rectifier current $I_{Ks}$) and *KCNH2* (responsible for the rapid delayed rectifier current $I_{Kr}$), respectively [47]. LQT1, in particular, is associated with sympathetically triggered cardiac events, specifically physical activity and water immersion [47]. During sympathetic stimulation, L-type calcium channel (LTCC) activity increases, resulting in a net influx of calcium ions and an increased propensity for early afterdepolarizations (EADs). *In vitro*, adrenergic stimulation provoked EADs in cardiomyocytes with KCNQ1 mutations derived from induced pluripotent stem cells (iPSCs), while β-blockers effectively prevented EADs [54]. EADs are typically caused by reactivation of LTCC during the plateau phase of the action potential, with reactivation facilitated by bradycardia, a hallmark of LQT1 from fetal life onwards [55].

Historically, children with severe LQTS, such as those with double LQT1 mutations, have often been misdiagnosed as epileptics [56], sometimes with tragic consequences as most antiepileptic drugs are contraindicated in LQTS. However, more recent studies have shown that the issue might be more complex, as a clinical seizure phenotype was found in 22% of genotype-ascertained LQT1 patients [4]. Moreover, dominant LQT1 mutations expressed in the heart and brain of mice, and corresponding to those linked with the genetic LQTS disorders Romano–Ward syndrome and Jervell and Lange-Nielsen syndrome in humans, resulted in both cardiac arrhythmias and epileptic seizures, revealing a potential dual neuro-cardiac arrhythmogenic role for *KCNQ1* mutations [3].

The synergy between both the CVS and the PNS in both CPVT and LQTS is further exemplified by the common preventive neuromodulator therapeutic strategies employed for the treatment of these disorders, which includes surgical cardiac sympathetic denervation [57–59]. Taken together, these observations highlight the need for more elaborate studies to elucidate the intricate mechanisms underlying neuro-cardiac transmission. Such studies will also have relevance for other diseases affecting the neuro-cardiac axis. Takotsubo syndrome (TTS), for example, a transient but severe form of acute heart failure [60], has been linked to sudden emotional or stressful events resulting in elevation in sympathetic activity

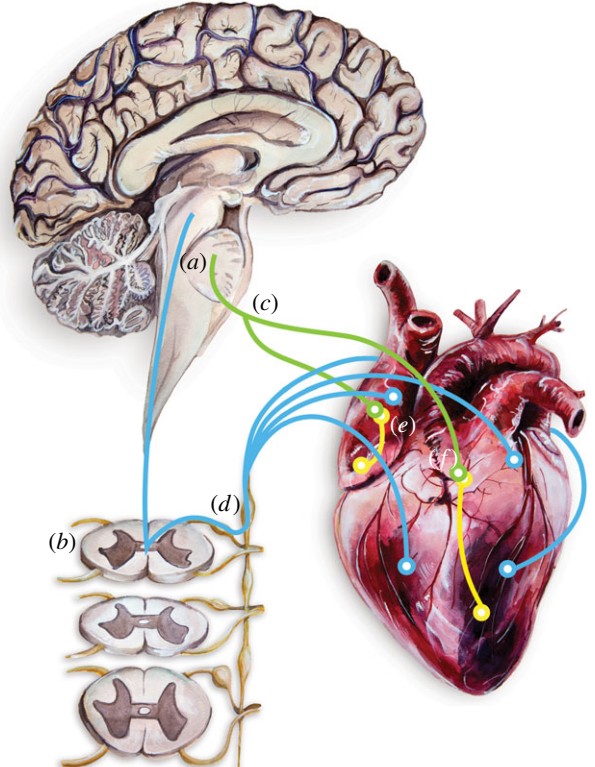

**Figure 1.** Simplified illustration of the centres of cardiac neurotransmission, where blue and green lines indicate potentially bidirectional pathways of both efferent and afferent communication. The brainstem (*a*) comprises the main autonomic centres of the CNS, parts of which may be stimulated through spreading depolarization during an epileptic seizure [14]. Neurons in these centres project via the spinal cord (*b*) or bilateral vagus nerves (*c*) to cardiac postganglionic neurons in the ganglia of the sympathetic chain (*d*) or the intrinsic ganglionated plexi on the heart (*e*), respectively (reviewed by Ashton *et al.* [15]). Ganglionated plexi neurons are situated predominantly in fat deposits on the surface of the heart (basal pale regions (*f*)), and they innervate the vasculature and tissues of the specialized conduction system and working myocardium. Painting and art work by M. Cremer.

[61–63]. Although the pathophysiology of TTS is not currently well understood [63], direct measurement in patients with TTS has shown an increase in sympathetic nerve activity associated with a decrease in spontaneous baroreflex control of sympathetic activity [64]. Studies that enable investigation of the modulation of cardiac tissue by sympathetic neurons at the cellular level are likely to be highly relevant for gaining a better understanding of the mechanisms that underlie such diseases.

## 3. *In vitro* approaches to examine neuro-cardiac interactions

While the need for more elaborate research on the pathophysiology of neuro-cardiac diseases is increasingly acknowledged [65,66], the complexity of neuro-cardiac interactions is difficult to investigate using whole animal models. For example, cross-talk between different centres of the afferent and efferent systems involved in cardiac neurotransmission (figure 1; more complex in-depth schemes are available such as [67]) can produce compensatory responses to pathological insults or drug effects, masking the underlying mechanisms that may be specific to either the neuronal or the cardiac tissue. Studies using engineered tissue culture models facilitate tighter control over the composition of the cell types (neuronal or cardiac) allowing for the investigation of tissue-specific responses induced by pathological factors (such as genetic mutations) [6] or drugs. As a result, it is possible to characterize the degree of influence of factors that are normally inaccessible using traditional *ex vivo* and *in vivo* techniques. When combined with advances in optical actuation and characterization, monolayer cultures are well suited for long-term, high-throughput investigation [28–31] of neuro-cardiac diseases. Here, we will discuss *in vitro* tissue culture approaches and recent advances in optical imaging of engineered tissue that could be applied to advance our current understanding of neuro-cardiac physiology and for the pathophysiology of neuro-cardiac diseases, such as CPVT.

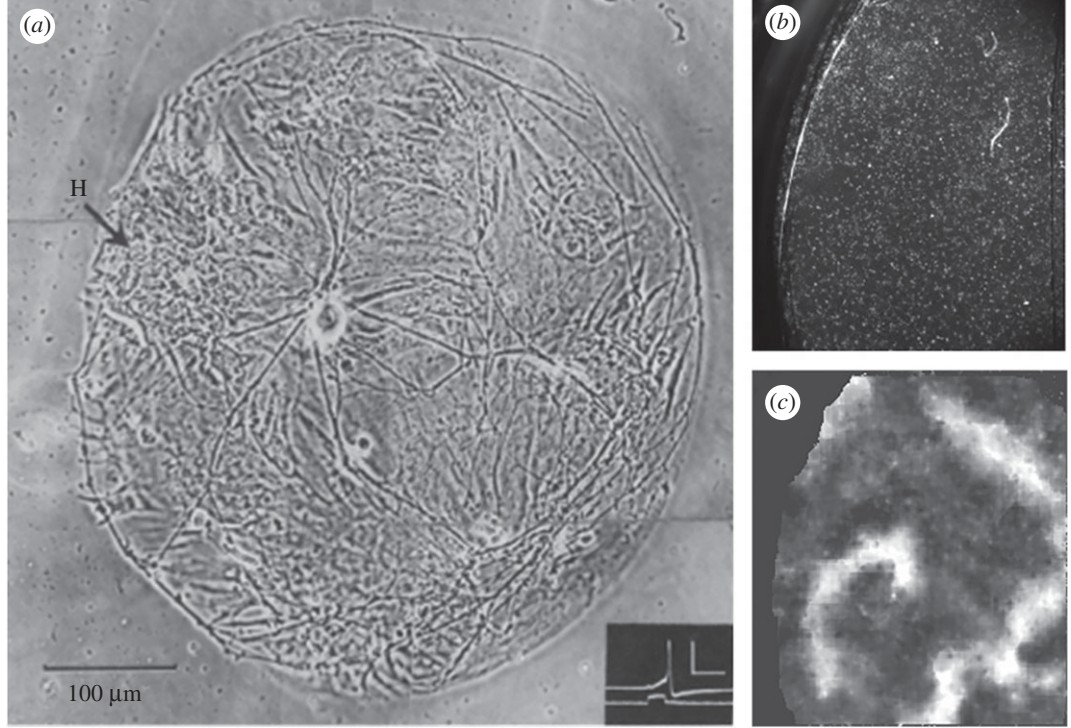

**Figure 2.** (*a*) Micro co-culture containing a solitary rat sympathetic neuron innervating cardiomyocyte clusters. A single cardiomyocyte cluster is indicated by the arrow at H. The neuron is at 19 days in culture. Inset shows a corresponding action potential recorded from this neuron in response to current injection. Inset scale represents 50 mV (*y*-axis) and 20 ms (*x*-axis). (*b,c*) Reconstruction of the wavefront in a cardiac monolayer. Representative raw image (*b*) and following background subtraction to reveal waves, a spiral pattern of excitation wavefront is observed (*c*) from a cardiac monolayer culture using dye-free optical imaging. (*a*) Reproduced from [68] with permission from Dr Peter R MacLeish, Morehouse School of Medicine, Atlanta, GA, USA, and Dr Paul O'Lague, Department of Molecular, Cell and Developmental Biology, University of California, Los Angeles, USA. (*b,c*) Adapted from [27] with permission from Dr Rebecca Burton and Prof. Gil Bub, University of Oxford.

## 3.1. Tissue engineering

Tissue engineering has proved to be beneficial in understanding neuro-cardiac pathophysiology. Co-culture techniques of neurons and myocytes, as demonstrated in the 1970s by Furshpan *et al.*, allow for cell-level access, providing an intermediate model between mono-culture and the multicellular complexity of whole organ experimentation (figure 2*a*) [68]. One method for preserving the extracellular matrix architecture and multicellular composition of native tissue is through human organotypic cultured cardiac slices as described by Kang *et al.* [26]. In mono-culture and co-culture systems, however, cell population composition, density and environment can be determined, thereby controlling factors out of reach in *ex vivo* and *in vivo* experiments. Altering the density of cell growth—sparse or confluent cultures—regulates the degree of cellular coupling by gap junctions [69]. For example, a confluent mono-culture of cardiomyocytes supports propagation of excitation waves [27,70] (figure 2*b,c*).

Alternatively, the sparse cardiomyocyte and sympathetic neuron cultures used by Furshpan *et al.* [68] reduced synaptic density, refining the connectivity between cultured cells (figure 2*a*, [68]). Tissue culture systems also confer precise control of the cellular environment through controlling the media composition and the culture surface. By culturing neurons in media conditioned by the growth of cardiomyocytes, Lockhart *et al.* [17] verified the importance of cell surface characteristics and hetero-cellular contact on neuron and cardiomyocyte differentiation [17]. The structural anisotropy of native cardiac tissues can be recapitulated through patterning of the culture surface, as shown by several studies [71–73]. Pattern analysis and spatial statistics have demonstrated that traditional neuronal cultures deviate from complete spatial randomness [74]. Patterned culture surfaces change the spatial organization of the somata and dendrites of cultured neurons enabling control of network morphology [74]. Structured patterning of myocytes and neurons offers a new model to address complex questions relating to neuro-cardiac behaviour in models of physiology and pathophysiology where tissue properties are directionally dependent.

## 3.2. Human-induced pluripotent stem cell tissue engineering approaches

Engineered tissue constructs support complex behaviours and recapitulate structure–function relationships enabling precise measurement of macroscopic behaviour (such as spiral re-entrant waves, figure 2c) [27,70,75–77]. The translational capacity of tissue culture models has improved with the design of anisotropic tissues as well as the use of human-induced pluripotent stem cells (hiPSCs) [54,72,73,78,79]. Species differences between rodents and humans can result in poor translation from *in vivo* experimental studies to human patients. Moretti *et al.* [54] and Itzhaki *et al.* [79] have demonstrated very elegantly the power and utility of induced pluripotent stem cells in recapitulating disease phenotype in LQTS [54,79]. However, the immaturity of the hiPSC phenotype presents challenges to their use for pathophysiological modelling and drug testing that should be taken into consideration in such studies [80–84]. For example, Jonsson *et al.* [80] have illustrated a lack of functional $I_{K1}$ potassium current and a shift in the activation threshold for sodium channel ($I_{Na}$) activation in human embryonic stem cell-derived cardiomyocytes which inhibited their ability to model arrhythmia in response to proarrhythmic drugs. The lack of functional $I_{K1}$ in hiPSCs should be of particular concern for studies aimed at modelling CPVT as mutations in KCNJ2, which encodes the pore-forming subunit of Kir2.1, have been linked to this disease [85]. The dynamic clamp [86] and optical dynamic clamp techniques [84] have, however, been used to simulate $I_{K1}$ behaviours in hiPSC, thus recapitulating human adult ventricular cardiomyocyte phenotypes and improving their utility for high-throughput screens. Robust functional coupling of hiPSC-derived neurons with target tissues is essential to study intercellular physiology, which has proven difficult to achieve. However, in 2016, Oh *et al.* [87] successfully demonstrated functional synapses between co-cultured hiPSC-derived sympathetic neurons and rat neonatal ventricular myocytes. While developing consistent hiPSC cardiac cultures with adult phenotypes is challenging [88,89], human tissue culture models of neuro-cardiac diseases provide the potential for high-throughput characterization, improving the translational success of drug research. Challenges remain in developing all human iPSC model systems with accurate adult phenotypes. Some of the problems to solve include the excessive variability observed in differentiated cells which includes discrepancies in genetic, epigenetic and transcriptional features and cardiomyocyte functionality [89].

## 3.3. Optical techniques in tissue culture

The planar surface topology of tissue culture is well suited to minimally invasive optical techniques, which allow for precision-controlled characterization of neuro-cardiac interactions. The curving surface, degree of movement, tissue thickness and inhomogeneity of *in vivo* and *ex vivo* cardiac tissues are challenging for optical characterization techniques [76]. Optical experimental techniques can be divided into two categories: optical actuation and optical characterization.

With optical actuation, light modulates cellular behaviour using techniques such as optogenetics and optically caged compounds [30,31,84,90–94]. Bruegmann *et al.* [93,94] have illustrated optogenetic actuation of *in vitro* and *in vivo* mouse cardiac tissues including defibrillation of ventricular tachycardia. Prando *et al.* used the spatio-temporal specificity of optogenetic actuation to characterize the dynamics of neuro-cardiac interactions *in vivo* as well as *in vitro* [66]. In culture, optogenetic actuation has been used to recover the $I_{K1}$ current in hiPSC [84] as well as induce and reverse spiral waves [27]. Since the kinetic and pharmacological characteristics of opsins are known to influence cellular behaviour (reviewed by O'Shea *et al.* [95]), researchers can control for these effects through tandem-cell-unit [28,96] and spatially controlled delivery patterns [97]. In the tandem-cell-unit method, opsins are expressed in non-excitable cells electrically coupled to cardiomyocytes leaving the electrophysiological properties of cardiomyocytes relatively intact [28,96]. Similarly, cell type-specific expression of opsins in sympathetic or parasympathetic neurons allows optical actuation of cardiac tissue by influencing the release of physiologically relevant mediators, such as norepinephrine or acetylcholine [98,99]. Transgene patterning has improved quantitative assessment of cell behaviour by tracking and specifically measuring transduced cells [97].

Optically caged compounds enable reversible localized release of biologically active molecules circumnavigating temporal delays and reducing the diffusion effects associated with washing-in a drug or compound [100]. Caged compounds are biologically active molecules made inert by a photoactive protecting group [91]. The spatial–temporal resolution of caged compounds is determined by the speed of a light pulse and kinetics of caging [91,95]. Typically, caged compounds enable instantaneous modification of intracellular signalling molecules.

Optical characterization techniques, such as fluorescent voltage- and calcium-sensitive dyes as well as Förster resonance energy transfer (FRET), are powerful tools for optically mapping cardiomyocyte physiology. High-resolution FRET imaging quantitatively assesses cellular processes at the resolution of microdomains through fluorescence shifts of a tagged biologically active compound [101]. The reversibility of these optical characterization techniques affords visualization of oscillatory cellular behaviours. For example, cAMP FRET sensors in co-culture have illuminated the structural importance of neuro-cardiac junctional domains [66], as well as potential pathogenesis of hypertension via the dominant role of sympathetic stellate neurons in driving β-adrenergic responses in an animal model of hypertension [6].

Optical mapping of conduction velocity in both whole hearts and engineered cardiac tissues is a useful tool for the study of cardiac arrhythmias, providing the ability to visualize in near real time the conduction of electrical signals. In culture, contact fluorescence imaging uses hexagonally packed optical fibres contacting the glass bottom of a culture dish to record electrically induced and electrically terminated arrhythmias [102]. An all-optical approach to actuation and characterization is possible with appropriate spectral spacing [28,96]. In 2016, Klimas et al. demonstrated an all-optical actuation and high-throughput characterization system called OptoDyCE [28]. However, phototoxicity and bleaching limit the functionality of fluorescence measurements for long-term characterization.

By contrast, dye-free optical imaging (figure 2b,c) circumvents the damaging heat and irradiance effects of fluorescent imaging by recording the rhythmic contractions of cardiomyocytes grown in monolayers using interference patterns generated by the interaction of light from an off-axis, partially coherent light source [27]. Non-invasive phase contrast macro-optics has been shown to help in visualizing the contractile motion of cardiac cells [75]. In 2004, Hwang et al. recorded complex re-entrant arrhythmias in cardiomyocyte mono-cultures by using propagation-induced changes in light intensity in phase contrast images [75]. Once the phase contrast data are processed for intensity variations with time, characteristic images of arrhythmic spiral waves emerge (figure 2c and also in [27,75,103]). Combining optical actuation with dye-free imaging, Burton et al. [27] demonstrated optical control over the chirality, induction and termination of spiral waves with patterned light. Dye-free imaging with optical actuation shows promise as a high-throughput method for co-culture studies and affords long-term monitoring (as opposed to fluorescence) and characterization data that is inaccessible with more traditional fluorescent imaging techniques. Traditional tools such as planar patch clamp systems and microelectrode arrays rely on contact-based interrogation [29]. As demonstrated by the dye-free, optical mapping studies aforementioned, all-optical approaches allow non-contact actuation with the precision needed to control wave patterns, which current electrical and pharmacological methods for wave modulation lack. These methods offer the opportunity to actively interrogate and manipulate cell dynamics in space and time (discussed in this review by Entcheva & Bub [104]). The data generated by this method can be used for analysing simple wave dynamics (for example, to calculate conduction velocity of planar waves). The interpretation of complex spiral waves or wavelet patterns may be hindered by similarities in the optical signals from excitation and relaxation waves (Supplementary fig. 1 in [27]).

# 4. Discussion and future directions

The complex pathophysiology of neuro-cardiac diseases is dependent upon an array of secretory factors, cell surface characteristics, rhythmic behaviours, connectivity and communication across multiple cell types including cardiomyocytes and neurons. Using techniques such as hetero-cellular tissue culture and micropatterned surfaces, experiments characterizing multisystem physiology can be designed with unprecedented reproducibility, specificity and cellular access. Tissue culture models provide experimental access to model the intricate connections between the PNS and CVS at the cellular level, a degree of access that is not possible using more traditional in vivo and ex vivo techniques. Cardiac tissue monolayers are excitable media that serve as an intermediary between in silico and in vivo experimentation [77,105,106]. Combining engineered tissue models with cellular electrophysiology, optical actuation and optical characterization further enhances the experimental precision accessible to characterize neuro-cardiac physiology and pathophysiology. Advances in combined optical actuation and characterization techniques present an opportunity for long-term, contact-free investigation of neuro-cardiac diseases. Further validation of tissue culture models and optical techniques present an opportunity for low-cost, high-throughput testing of therapeutics for complex neuro-cardiac diseases.

Data accessibility. This article has no additional data.

Authors' contributions. R.A.B.B. initiated the review topic. R.A.B.B., C.S. and M.C. designed the review. M.C. painted the art work in figure 1. All authors contributed to the writing, editing and approval of the manuscript.

Competing interests. The authors have nothing to declare.

Funding. This work was supported by a Colin Pillinger Royal Society International Exchange Award (IE160164). R.A.B.B. is funded by a Sir Henry Dale Wellcome Trust and Royal Society Fellowship (grant no. 109371/Z/15/Z) and acknowledges support from The Returning Carers' Fund, Medical Sciences Division, University of Oxford. R.A.B.B. is a Senior Research Fellow at Linacre College. S.J.B. is funded by a British Heart Foundation Project Grant (PG/18/4/33521). C.S. is funded by the Blaschko Trust (Department of Pharmacology and Linacre College, University of Oxford). C.S. is a Junior Research Fellow at Linacre College. J.M.M. acknowledges support from the Freemasons Foundation, Marsden Fund (18-UoA-100), and Auckland Medical Research Foundation (1118003). A.W. is funded by the Hugh Green Foundation, Cure Kids and the Auckland Medical Research Foundation. G.B. acknowledges support from the Canadian Heart and Stroke Foundation.

Acknowledgements. We thank Prof. Neil Herring, University of Oxford for comments and suggestions relating to figure 1.

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
