## [Reviewer comments · Royal Society Open Science]

Review History

RSOS-200265.R0 (Original submission)

Review form: Reviewer 1

Is the manuscript scientifically sound in its present form?

Yes

Are the interpretations and conclusions justified by the results?

Yes

Is the language acceptable?

Yes

Do you have any ethical concerns with this paper?

No

Have you any concerns about statistical analyses in this paper?

No

Recommendation?

Accept with minor revision (please list in comments)

Comments to the Author(s)

The manuscript submitted by the group of Professor Burton provides a timely short review of experimental options for studying the interaction between neurons and cardiomyocytes, and vice versa. The manuscript reads well and it is my expectation it will be of interest to the readership of RSOS. There are a few aspects that I think should be covered in more depth in order to further improve the manuscript.

1. In vitro models consisting of human stem cell-derived cells are an attractive option because of their human origin, however the immature phenotype of (at least) hiPSC-CM is not discussed. To make the review more balanced, this topic has to be discussed to my opinion. In the context of safety pharmacology, this aspect has been covered quite a bit, and the same concerns should also apply here (see e.g. PMIDs 28986101, 22353256, 28686863). Similar concerns are likely to apply to hiPSC-derived neurons.

2. Related to the immaturity of hiPSC-CM is lack of expression of IK1 channels, encoded by the KCNJ2 gene. In section B, the authors discuss CPVT as one of the diseases in which the neuro-cardiac axis is an important aspect of pathophysiology. While CPVT is mostly associated with aberrant calcium handling, also mutations in the KCNJ2 gene have been implied (PMID 19843922). This may be particularly relevant when trying to model CPVT in a tissue construct containing immature cardiomyocytes that have a lack of IK1 channels.

3. Section C3 discusses the use of dye-free optical imaging as a method for long-term characterisation. Clearly, contraction based data may miss some of the subtleties of the interaction between neurons and cardiomyocytes. Which are the strong and weak parts of this approach compared to e.g. fluorescence or MEA techniques?

Minor points:

1. Section C1 includes a reference to Kang et al., 2016. This is missing from the list of citations.
2. Page 9, first line, discusses the influence of cell density in a culture on cellular coupling. Do the authors mean gap junctional conductance, or likelihood that a cell is coupled to a neighbouring cell?
3. Section C2 concludes that culture models "provide the potential for high-throughput characterisation", but which are the problems left to solve, please elaborate a bit.

Review form: Reviewer 2

Is the manuscript scientifically sound in its present form?

Yes

Are the interpretations and conclusions justified by the results?

Yes

Is the language acceptable?

Yes

Do you have any ethical concerns with this paper?

No

Have you any concerns about statistical analyses in this paper?

No

Recommendation?

Accept with minor revision (please list in comments)

Comments to the Author(s)

The paper "Combining tissue engineering and optical imaging approaches to explore interactions along the neuro-cardiac axis" provides a review of the role of engineered tissue in the study of diseases that span cardiology and neurology. I found the paper interesting, timely and well written.

The granularity of my comments reflects that all other aspects of the paper were acceptable.

In some places the text made large logical or thematic leaps and this could be improved to make the paper easier to read. A few sections seemed uncoupled from the next.

There is a large number of authors for a short review.

Minor: Line 13-17 this sentence does not make a point, just an observation. It is not clear if this implies that the same trigger that causes the seizure causes the cardiac event, if the stress placed on the body initiates the cardiac event or if the cardiac event is due to mis firing neurological signals. Explaining the take home point from this sentence would support the paragraph conclusion.

Line 22-24 is not clear if the patient's had epilepsy as well as CPVT or were simply misdiagnosed. The number of cases that this represents should be provided as if this is low I do not believe that a strong /any conclusion can be drawn from a few misdiagnosis.

The conclusion of paragraph 1 is not well motivated by the preceding text.

Page 5 line 18: A little bit more of a transition from the problem definition to the review motivation would be useful. What are the current methods for studying cardiac-neuro? Why are these insufficient?

Page 5 Line 52: There is no clear transition to this section, it is not related to the clinical problem described above, nor is it related to engineered tissue or optical imaging. It may be useful to have something about Neuro-Cardiac diseases acting across multiple scales where problems in the cell emerge integrate into pathologies at the organ scale.

Page 6 paragraph 1. There are 5 paragraphs in this section. 2 on CPVT (calcium related) and 2 on LQTS (ion channel related). This paragraph focuses on calcium with no mention of transmembrane ion channels. A more balanced introduction, possibly flagging the two subsequent diseases that they relate to would be nice.

Page 8 section C is the motivation for this review/method. It would be nice to see more than one paper supporting this position cited and possible a paper that was not from Oxford. To demonstrate this is a generally held position.

Page 10 line 24. (Bub and Burton 2015, Burton et al., 2015) Surely someone else has also done this type of work

The paper could have mentioned Takotsubo syndrome
I was surprised by the absence of any mention of optical genetics.

Decision letter (RSOS-200265.R0)

Dear Dr Burton

On behalf of the Editors, I am pleased to inform you that your Manuscript RSOS-200265 entitled "Combining tissue engineering and optical imaging approaches to explore interactions along the neuro-cardiac axis" has been accepted for publication in Royal Society Open Science subject to minor revision in accordance with the referee suggestions. Please find the referees' comments at the end of this email.

The reviewers and handling editors have recommended publication, but also suggest some minor revisions to your manuscript. Therefore, I invite you to respond to the comments and revise your manuscript.

- Ethics statement

- Data accessibility

<http://datadryad.org/submit?journalID=RSOS&manu=RSOS-200265>

- Competing interests

- Authors' contributions

- Acknowledgements

- Funding statement

Because the schedule for publication is very tight, it is a condition of publication that you submit the revised version of your manuscript before 06-May-2020. Please note that the revision deadline will expire at 00.00am on this date. If you do not think you will be able to meet this date please let me know immediately.

- 1) A text file of the manuscript (tex, txt, rtf, docx or doc), references, tables (including captions) and figure captions. Do not upload a PDF as your "Main Document";
- 2) A separate electronic file of each figure (EPS or print-quality PDF preferred (either format should be produced directly from original creation package), or original software format);
- 3) Included a 100 word media summary of your paper when requested at submission. Please ensure you have entered correct contact details (email, institution and telephone) in your user account;
- 4) Included the raw data to support the claims made in your paper. You can either include your data as electronic supplementary material or upload to a repository and include the relevant doi within your manuscript. Make sure it is clear in your data accessibility statement how the data can be accessed;

5) All supplementary materials accompanying an accepted article will be treated as in their final form. Note that the Royal Society will neither edit nor typeset supplementary material and it will be hosted as provided. Please ensure that the supplementary material includes the paper details where possible (authors, article title, journal name).

If your manuscript is newly submitted and subsequently accepted for publication, you will be asked to pay the article processing charge, unless you request a waiver and this is approved by Royal Society Publishing. You can find out more about the charges at <https://royalsocietypublishing.org/rsos/charges>. Should you have any queries, please contact opscience@royalsociety.org.

Royal Society Open Science
opscience@royalsociety.org

on behalf of Dr Xiaoyu Luo (Associate Editor) and Pietro Cicuta (Subject Editor)
opscience@royalsociety.org

Associate Editor Comments to Author (Dr Xiaoyu Luo):

Comments to the Author:
Accept with minor revisions

Reviewer comments to Author:
Reviewer: 1

Comments to the Author(s)

The manuscript submitted by the group of Professor Burton provides a timely short review of experimental options for studying the interaction between neurons and cardiomyocytes, and vice versa. The manuscript reads well and it is my expectation it will be of interest to the readership of RSOS. There are a few aspects that I think should be covered in more depth in order to further improve the manuscript.

1. In vitro models consisting of human stem cell-derived cells are an attractive option because of

their human origin, however the immature phenotype of (at least) hiPSC-CM is not discussed. To make the review more balanced, this topic has to be discussed to my opinion. In the context of safety pharmacology, this aspect has been covered quite a bit, and the same concerns should also apply here (see e.g. PMIDs 28986101, 22353256, 28686863). Similar concerns are likely to apply to hiPSC-derived neurons.

2. Related to the immaturity of hiPSC-CM is lack of expression of IK1 channels, encoded by the KCNJ2 gene. In section B, the authors discuss CPVT as one of the diseases in which the neuro-cardiac axis is an important aspect of pathophysiology. While CPVT is mostly associated with aberrant calcium handling, also mutations in the KCNJ2 gene have been implied (PMID 19843922). This may be particularly relevant when trying to model CPVT in a tissue construct containing immature cardiomyocytes that have a lack of IK1 channels.

3. Section C3 discusses the use of dye-free optical imaging as a method for long-term characterisation. Clearly, contraction based data may miss some of the subtleties of the interaction between neurons and cardiomyocytes. Which are the strong and weak parts of this approach compared to e.g. fluorescence or MEA techniques?

Minor points:

1. Section C1 includes a reference to Kang et al., 2016. This is missing from the list of citations.
2. Page 9, first line, discusses the influence of cell density in a culture on cellular coupling. Do the authors mean gap junctional conductance, or likelihood that a cell is coupled to a neighbouring cell?
3. Section C2 concludes that culture models "provide the potential for high-throughput characterisation", but which are the problems left to solve, please elaborate a bit.

Reviewer: 2

Comments to the Author(s)

The paper "Combining tissue engineering and optical imaging approaches to explore interactions along the neuro-cardiac axis" provides a review of the role of engineered tissue in the study of diseases that span cardiology and neurology. I found the paper interesting, timely and well written.

The granularity of my comments reflects that all other aspects of the paper were acceptable.

In some places the text made large logical or thematic leaps and this could be improved to make the paper easier to read. A few sections seemed uncoupled from the next.

There is a large number of authors for a short review.

Minor: Line 13-17 this sentence does not make a point, just an observation. It is not clear if this implies that the same trigger that causes the seizure causes the cardiac event, if the stress placed on the body initiates the cardiac event or if the cardiac event is due to mis firing neurological signals. Explaining the take home point from this sentence would support the paragraph conclusion.

Line 22-24 is not clear if the patient's had epilepsy as well as CPVT or were simply misdiagnosed. The number of cases that this represents should be provided as if this is low I do not believe that a strong /any conclusion can be drawn from a few misdiagnosis.

The conclusion of paragraph 1 is not well motivated by the preceding text.

Page 5 line 18: A little bit more of a transition from the problem definition to the review

motivation would be useful. What are the current methods for studying cardiac-neuro? Why are these insufficient?

Page 5 Line 52: There is no clear transition to this section, it is not related to the clinical problem described above, nor is it related to engineered tissue or optical imaging. It may be useful to have something about Neuro-Cardiac diseases acting across multiple scales where problems in the cell emerge integrate into pathologies at the organ scale.

Page 6 paragraph 1. There are 5 paragraphs in this section. 2 on CPVT (calcium related) and 2 on LQTS (ion channel related). This paragraph focuses on calcium with no mention of transmembrane ion channels. A more balanced introduction, possibly flagging the two subsequent diseases that they relate to would be nice.

Page 8 section C is the motivation for this review/method. It would be nice to see more than one paper supporting this position cited and possible a paper that was not from Oxford. To demonstrate this is a generally held position.

Page 10 line 24. (Bub and Burton 2015, Burton et al., 2015) Surely someone else has also done this type of work

The paper could have mentioned Takotsubo syndrome
I was surprised by the absence of any mention of optical genetics.

Author's Response to Decision Letter for (RSOS-200265.R0)

See Appendix A.

Decision letter (RSOS-200265.R1)

Dear Dr Burton,

It is a pleasure to accept your manuscript entitled "Combining tissue engineering and optical imaging approaches to explore interactions along the neuro-cardiac axis" in its current form for publication in Royal Society Open Science.

Kind regards,

on behalf of Dr Xiaoyu Luo (Associate Editor) and Pietro Cicuta (Subject Editor)
openscience@royalsociety.org

Associate Editor Comments to Author (Dr Xiaoyu Luo):

The authors have addressed all the minor issues of the reviewers.

Appendix A

We would like to thank the reviewers for their helpful comments and suggestions. Below we detail the changes made to the manuscript to address these. Please follow the tracked document to note all the changes mentioned below. Thanks.

Reviewer comments to Author:

Reviewer: 1

Comments to the Author(s)

The manuscript submitted by the group of Professor Burton provides a timely short review of experimental options for studying the interaction between neurons and cardiomyocytes, and vice versa. The manuscript reads well and it is my expectation it will be of interest to the readership of RSOS. There are a few aspects that I think should be covered in more depth in order to further improve the manuscript.

1. In vitro models consisting of human stem cell-derived cells are an attractive option because of their human origin, however the immature phenotype of (at least) hiPSC-CM is not discussed. To make the review more balanced, this topic has to be discussed to my opinion. In the context of safety pharmacology, this aspect has been covered quite a bit, and the same concerns should also apply here (see e.g. PMIDs 28986101, 22353256, 28686863). Similar concerns are likely to apply to hiPSC-derived neurons.

Answer: We thank the reviewer for drawing the lack of discussion of the hiPSC immature phenotype in this manuscript. Discussion of these points has been added to the text on page 10.

2. Related to the immaturity of hiPSC-CM is lack of expression of IK1 channels, encoded by the KCNJ2 gene. In section B, the authors discuss CPVT as one of the diseases in which the neuro-cardiac axis is an important aspect of pathophysiology. While CPVT is mostly associated with aberrant calcium handling, also mutations in the KCNJ2 gene have been implied (PMID 19843922). This may be particularly relevant when trying to model CPVT in a tissue construct containing immature cardiomyocytes that have a lack of IK1 channels.

Answer: We have now highlighted the importance of KCNJ2 mutations. Discussion of this point and references have been added to the text on page 10.

3. Section C3 discusses the use of dye-free optical imaging as a method for long-term characterisation. Clearly, contraction based data may miss some of the subtleties of the interaction between neurons and cardiomyocytes. Which are the strong and weak parts of this approach compared to e.g. fluorescence or MEA techniques?

Answer: We have incorporated new text to discuss this on page 12-13 and highlight a review that looks into the merits, strengths and limitations of all optical interrogation control of cardiac excitation combined with high-resolution optogenetic actuation and optical mapping.

Minor points:

1. Section C1 includes a reference to Kang et al., 2016. This is missing from the list of citations.

Answer: We have now added the missing citation.

2. Page 9, first line, discusses the influence of cell density in a culture on cellular coupling. Do the authors mean gap junctional conductance, or likelihood that a cell is coupled to a neighbouring cell?

Answer: We have added an explanation in the text that this influence is through gap junctions.

3. Section C2 concludes that culture models "provide the potential for high-throughput characterisation", but which are the problems left to solve, please elaborate a bit.

Answer: We have added further details on this point, page 10.

Reviewer: 2

Comments to the Author(s)

The paper "Combining tissue engineering and optical imaging approaches to explore interactions along the neuro-cardiac axis" provides a review of the role of engineered tissue in the study of diseases that span cardiology and neurology. I found the paper interesting, timely and well written.

The granularity of my comments reflects that all other aspects of the paper were acceptable.

In some places the text made large logical or thematic leaps and this could be improved to make the paper easier to read. A few sections seemed uncoupled from the next.

There is a large number of authors for a short review.

Minor: Line 13-17 this sentence does not make a point, just an observation. It is not clear if this implies that the same trigger that causes the seizure causes the cardiac event, if the stress placed on the body initiates the cardiac event or if the cardiac event is due to mis firing neurological signals. Explaining the take home point from this sentence would support the paragraph conclusion.

Answer: We have now modified the text accordingly to make clear that the answers to these questions remain to be answered by further research (page 2).

Line 22-24 is not clear if the patient's had epilepsy as well as CPVT or were simply misdiagnosed. The number of cases that this represents should be provided as if this is low I do not believe that a strong /any conclusion can be drawn from a few misdiagnosis.

Answer: We have now described that seizures have also been reported in up to 50% of CPVT patients earlier in life before the presentation of CVPT-related symptoms (first paragraph page 2).

The conclusion of paragraph 1 is not well motivated by the preceding text.

Answer: The modifications we have made in paragraph 1 now provide a clearer motivation for the conclusion (page 2).

Page 5 line 18: A little bit more of a transition from the problem definition to the review motivation would be useful. What are the current methods for studying cardiac-neuro? Why are these insufficient?

Answer: We have now added a new paragraph describing the current methods in the study of neurocardiac interactions (page 3,4). Last paragraph on page 12-13 also discusses traditional fluorescent techniques, electrophysiology (MEA) and current methods.

Page 5 Line 52: There is no clear transition to this section, it is not related to the clinical problem described above, nor is it related to engineered tissue or optical imaging. It may be useful to have something about Neuro-Cardiac diseases acting across multiple scales where problems in the cell emerge integrate into pathologies at the organ scale.

Answer: We have modified the last paragraph of the introduction to link the methods we describe with the common neural and cardiac mechanisms involved in the pathophysiology of neurocardiac disease (page 3). Also the authors believe that the concept of 'problems emerging in cells can integrate into pathologies at the organ scale' is a general principle in diseases that is not required to be described in this short review.

Page 6 paragraph 1. There are 5 paragraphs in this section. 2 on CPVT (calcium related) and 2 on LQTS (ion channel related). This paragraph focuses on calcium with no mention of transmembrane ion channels. A more balanced introduction, possibly flagging the two subsequent diseases that they relate to would be nice.

Answer: Both CPVT and LQTS are related to impaired function of transmembrane ion channels. As explained in the second paragraph of Section B, RyR2 is the main **calcium release channel** involved in calcium induced calcium release during excitation-contraction (EC) coupling in cardiac cells. To avoid shifting the focus on Ca only, we have now added the following sentence 'To illustrate the importance of such methods, we will first discuss how common cellular mechanisms, such as Ca²⁺ or K⁺ regulation through transmembrane channels, can affect both neural and cardiac tissue in CPVT and LQTS, as examples of diseases that were initially considered solely cardiac..' in the paragraph preceding paragraph

1 in page 6. The underlined text states that we focus on two types of ion channel regulation in neurocardiac diseases.

Page 8 section C is the motivation for this review/method. It would be nice to see more than one paper supporting this position cited and possible a paper that was not from Oxford. To demonstrate this is a generally held position.

Answer: Unfortunately, these techniques have not been used extensively, highlighting the need to draw the reader's attention to the availability of such non-dye recording techniques. The original submission did however make mention of a study by Hwang *et al* (2004) from Korea University where the value of these recordings has also been demonstrated and we have attempted to highlight this work in addition to that from Oxford. We introduce a review on page 13 to help with this point raised (<https://www.ncbi.nlm.nih.gov/pmc/articles/PMC4850200/>).

Page 10 line 24. (Bub and Burton 2015, Burton et al., 2015) Surely someone else has also done this type of work

Answer: We have added the following sentence 'Non-invasive phase contrast macro-optics has been shown to help in visualising the contractile motion of cardiac cells (Hwang et al., 2004).' (page 12)

The paper could have mentioned Takotsubo syndrome

We thank the reviewer for drawing our attention to the relevance of Takotsubo syndrome to these studies. We have now added a mention of this syndrome and its relevance to the end of Section B on page 6-7.

I was surprised by the absence of any mention of optical genetics.

Answer: We have described optogenetics in section paragraph 2 of C3. We have now added more examples of studies using optogenetics (page 11).